# TraceDet: Hallucination Detection from the Decoding Trace of Diffusion Large Language Models

**Shenxu Chang**[1,*] **Junchi Yu**[1,*] **Weixing Wang**[2]**, Yongqiang Chen**[3,4]**, Jialin Yu**[1]
**Philip Torr**[1]**, Jindong Gu**[1]

[1]Department of Engineering Science, University of Oxford, UK
[2]Hasso Plattner Institute, University of Potsdam
[3]Carnegie Mellon University
[4]Mohamed bin Zayed University of Artificial Intelligence

`shenxu.chang@hertford.ox.ac.uk, junchi.yu@eng.ox.ac.uk,`
`weixing.wang@hpi.de, yqchen24@gmail.com,`
`jialin.yu@eng.ox.ac.uk philip.torr@eng.ox.ac.uk, jindong.gu@eng.ox.ac.uk`

## Abstract

Diffusion large language models (D-LLMs) have recently emerged as a promising alternative to auto-regressive LLMs (AR-LLMs). However, the hallucination problem in D-LLMs remains underexplored, limiting their reliability in real-world applications. Existing hallucination detection methods are designed for AR-LLMs and rely on signals from *single-step* generation, making them ill-suited for D-LLMs where hallucination signals often emerge throughout the *multi-step* denoising process. To bridge this gap, we propose **TraceDet**, a novel framework that explicitly leverages the intermediate denoising steps of D-LLMs for hallucination detection. TraceDet models the denoising process as an *action trace*, with each action defined as the model's prediction over the cleaned response, conditioned on the previous intermediate output. By identifying the sub-trace that is maximally informative to the hallucinated responses, TraceDet leverages the key hallucination signals in the multi-step denoising process of D-LLMs for hallucination detection. Extensive experiments on various open source D-LLMs demonstrate that **TraceDet** consistently improves hallucination detection, achieving an average gain in AUROC of 15.2% compared to baselines. Code is now available at `https://github.com/chang-sx/TraceDet`.

## 1 Introduction

The auto-regressive large language models (AR-LLMs) (Achiam et al., 2023; Vaswani et al., 2017) have demonstrated unprecedented capabilities in content generation (Maleki & Zhao, 2024) and general task completion (Yao et al., 2023). Despite their success, AR-LLMs still face challenges related to generation efficiency and the reversal curse due to the inherent limitation of the next-token prediction paradigm (Bachmann & Nagarajan, 2024). The diffusion large language models (D-LLMs) have emerged as a promising alternative to AR-LLMs. Unlike AR-LLMs that generate language sequences from left to right, D-LLMs iteratively denoise the whole language sequences with a bi-directional attention architecture. Thus, D-LLMs have great potential in efficient computation and more flexible reasoning. Recent open-sourced works, such as LLaDA and Dream model series (Nie et al., 2025; Ye et al., 2025), have successfully scaled D-LLMs to 8B parameters, achieving performance comparable to leading AR-LLMs (AI, 2024) at the same scale in various tasks.

Although most work focuses on enhancing the capability of D-LLMs (Zhao et al., 2025; Yang et al., 2025b), less focus is devoted to their hallucination problem. Hallucination refers to generating linguistically plausible yet factually incorrect contents, which is recognized as a byproduct of the

---

*Equal Contribution

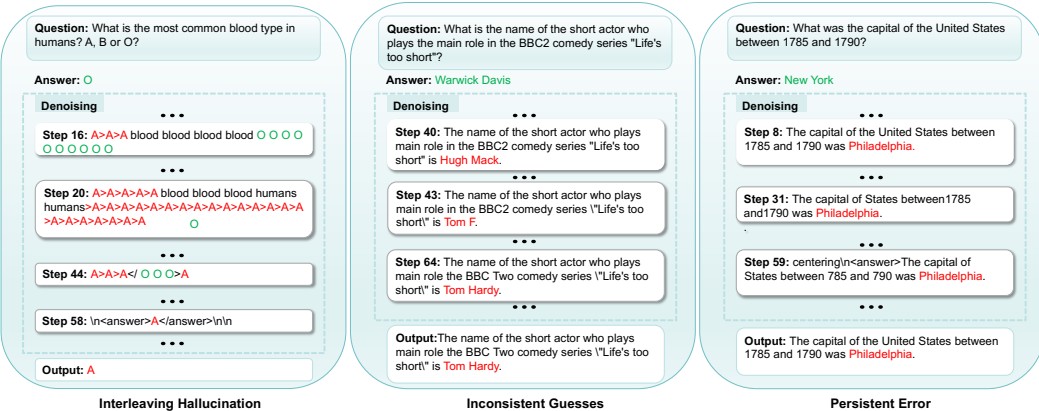

Figure 1: Illustration of representative D-LLM hallucination patterns extracted by TraceDet. Left: **Interleaving Hallucination**, where the model decodes both truthful and hallucinated content. Middle: **Inconsistent Guesses**, where multiple contradictory keywords lead to hallucination. Right: **Persistent Error**, where the model maintains a hallucinated answer throughout denoising. Hallucinations are highlighted with red.

increasing capability of language models (Manakul et al., 2023; Zhang et al., 2025). The hallucination issue in D-LLMs undermines user trust and potentially causes severe consequences in critical domains (Huang et al., 2025), hindering their deployment in safety-critical scenarios.

Existing literature has focused on hallucination detection in AR-LLMs, which can be broadly categorized into output-based detection (Kossen et al., 2024) and latent-based detection (Du et al., 2024). Output-based detection leverages hallucination-related signals derived from model outputs, such as the consistency across multiple sampled responses (Kuhn et al., 2023) or the entropy of token-level logits. The intuition is that hallucinated responses are typically associated with lower confidence (Rawte et al., 2023). Latent-based methods instead probe the hidden representations of AR-LLMs during a single forward pass to distinguish hallucinated from factual responses (Park et al., 2025; Li et al., 2025). Recent works further introduce new techniques to enhance the separation between hallucinated and factual responses in the latent space (Liu et al., 2025b; Orgad et al., 2025).

However, existing hallucination detection methods face challenges in detecting hallucinations in D-LLMs, since they typically exploit hallucination signals in the *single-step* generation process of AR-LLMs. Unlike AR-LLMs that produce responses in a single forward pass, D-LLMs iteratively refine the responses through a *multi-step* denoising process (Sahoo et al., 2024; Shi et al., 2024). Our empirical observations show that the hallucinated responses in D-LLMs are associated with an intriguing denoising process. As shown in Figure 1, some of them oscillate between factual and hallucinated content or randomly guess among various hallucinated answers, whereas others persistently maintain a single hallucinated answer throughout the denoising trajectory. While the underlying mechanism behind these behaviors remains an open question, these intermediate dynamics provide valuable signals for hallucination detection in D-LLMs.

**Proposed work**. We introduce **TraceDet**, a novel framework that leverages the intermediate denoising steps of D-LLMs for hallucination detection. The key insight is to represent the denoising process as an *action trace* (Black et al., 2024), where each action corresponds to the model's prediction of a complete response sequence given the intermediate result at one denoising step. Rather than relying on the final output, TraceDet aims to identify a sub-trace of the whole action trace that contributes to the hallucinated responses (Section 3.2). The major difficulty is that the sub-trace of actions is not known *a priori*, leading to the absence of explicit action labels for supervision. Inspired by the information-bottleneck (IB) principle (Tishby et al., 2000), TraceDet identifies the sub-trace of actions that are maximally informative to the hallucinated response (Section 3.3). This informative sub-trace is then used to train a classifier for final hallucination detection (Section 3.4).

We extensively evaluate the capability of the proposed TraceDet on the two available open-source D-LLMs, including LLaDA-8B-Instruct and Dream-7B-Instruct, across three QA datasets covering multiple-choice, open-ended, and contextual answering tasks. On average, TraceDet delivers a consistent improvement of **15.2%** in hallucination detection accuracy (AUROC), and further

studies also confirm the robustness of the proposed method to varying denoising strategies and hyperparameter settings. Our main contributions are threefold:

- We make an initial effort in the study of hallucination behaviors in D-LLMs, uncovering distinctive multi-step patterns, such as interleaving hallucination, inconsistent guesses, and persistent errors, that are absent in AR-LLMs.

- We introduce **TraceDet**, a novel hallucination detection framework that formulates the D-LLM denoising process as an action trace. By applying the information bottleneck principle, TraceDet automatically extracts the most informative sub-trace for detection, without requiring explicit step-level supervision.

- We conduct comprehensive experiments on two open-source D-LLMs (LLaDA-8B-Instruct, Dream-7B-Instruct) across diverse QA benchmarks, where our method shows an average AUROC improvement of $15.2\%$ over the baselines and demonstrates robustness across different denoising strategies and hyperparameter choices.

## 2 RELATED WORK

**Hallucination Detection** is a central problem in ensuring the safety, truthfulness, and faithfulness of LLM-based applications (Huang et al., 2025). Existing works have mainly been developed for AR-LLMs, and can be categorized as: (a) *Output-based:* calculating measures based on output signals such as semantic entropy (Kuhn et al., 2023), lexical similarity (Lin et al., 2023) and so on (Ren et al., 2022; Malinin & Gales, 2020; Xiong et al., 2023; Lin et al., 2022; Kuhn et al., 2023; Manakul et al., 2023); (b) *Latent-based:* probing hidden states from one-pass generations (Azaria & Mitchell, 2023; Chen et al., 2024a; Du et al., 2024; Su et al., 2024; Chen et al., 2024b; Li et al., 2023; Kossen et al., 2024; Marks & Tegmark, 2023; Kim et al., 2024; Chen et al., 2024a). While effective in AR-LLMs, existing methods face challenges in D-LLMs due to mismatches between final outputs and the intermediate denoising process, as well as the restricted availability of output token logits. We discuss the most relevant literature with our work, and defer the details to Appendix B.

**Diffusion Large Language Model** extends the success of diffusion models (Yang et al., 2023; Lu et al., 2025) to texts (Li et al., 2022). Nie et al. (2025) adopts an alternative discrete feature unifying the discrete remasking process, and has successfully scaled the D-LLM up to an 8B parameter level, achieving performance comparable to leading LLMs such as LLaMA-3 (AI, 2024). In addition, Dream-7B (Ye et al., 2025) adopts the same configurations as Qwen2.5-7B (Yang et al., 2025a) trained under a diffusion paradigm. Despite these advances, the hallucination problem in D-LLMs remains underexplored, hindering their application in real-world scenarios.

**Information Bottleneck** principle was initially proposed in signal processing to extract the most informative sub-instances while minimizing irrelevant information, and has gained lots of applications in deep learning (Alemi et al., 2016; West et al., 2019; Zhu et al., 2024; Luo et al., 2019). However, its use in hallucination detection remains largely unexplored due to the limited information available in a single text generation. The most related work (Bai et al., 2025) integrates IB into VLLM as a sub-instance extractor for images, mitigating hallucination in VLLM outputs. Our work builds on the sequential and information-excess nature of stepwise D-LLM generations. We propose a temporal embedding approach that captures intermediate step signals to detect hallucinations.

## 3 HALLUCINATION DETECTION IN DIFFUSION LLMS

### 3.1 PROBLEM FORMULATION

Unlike AR-LLMs, D-LLMs generate responses by iterative refinement through a forward noising and backward denoising process over $T$ time steps. Let $r = (r_0, \ldots, r_T)$ denote the sequence of intermediate texts, where each $r_t \in \mathbf{V}^n$ is a token sequence of length $n$ and $r_t^i \in \mathbf{V}$ for $i \in \{1, \ldots, n\}$. Here, $r_0$ is the clean text sequence and $r_T$ the fully masked sequence, with $t \in \{0, \ldots, T\}$. Given a prompt $p_0$, the forward noising process is defined by a sequence of distributions $\{q(r_t \mid r_{t-1})\}_{t=1}^T$, and can be written as $q(r_{1:T} \mid r_0) = \prod_{t=1}^T q(r_t \mid r_{t-1})$.

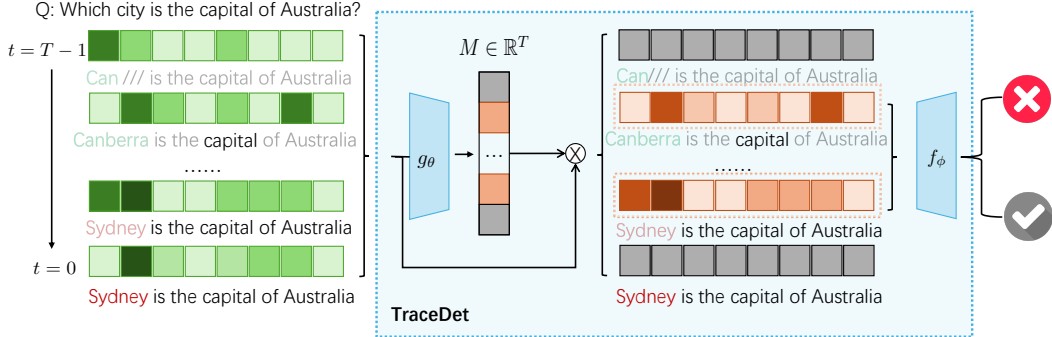

Figure 2: Illustration of TRACEDET. During denoising, a diffusion LLM generates intermediate sequences along with token-level entropy traces, where highlighted words indicate the retained tokens after remasking (left). The sub-instance extractor $g_\theta$ produces a temporal mask $M$ to focus on informative steps, and the predictor $f_\phi$ classifies whether the final response is hallucinated (right).

The reverse denoising process with discrete remasking is parameterized by the sequence of conditional distributions $\{P_\theta(r_0 \mid r_t, p_0)\}_{t=1}^T$. At each timestep $t > 0$, the model predicts all masked tokens from $r_t$: $\tilde{r}_{t-1} \sim P_\theta(r_0 \mid r_t, p_0)$, and then a fraction $\rho_t$ of tokens in $r_t$ are remasked to form $r_{t-1}$. As proposed in Nie et al. (2025), current popular remasking strategies often use low-confidence remasking, which retains the highest confidence tokens during each remasking step. After $T$ iterations, the process outputs the final response $r_0$.

Given an input query $p_0$, the hallucination detection can be formulated as binary classification:

$$\min_{h \in \mathcal{H}} \mathcal{L}(Y, h(r_0)), \quad s.t. \quad r_0 \sim \prod_{t=T-1}^{0} P_\theta(r_t \mid r_{t+1}, p_0), \tag{1}$$

where $r_0$ is the final response from the D-LLM, which can be either a hallucinated or non-hallucinated (or factual) answer, where $Y \in \{0, 1\}$ is the ground-truth label indicating if $r_0$ is a hallucinated answer, $h$ is the classifier belonging to some hypothesis space $\mathcal{H}$ and $\mathcal{L}(\cdot)$ is the cross-entropy loss, and $r_{T-i}$ is the intermediate sequence produced at the $i$-th denoising step. Since hallucination detection directly on generated text is costly and difficult to implement in practice, we instead rely on auxiliary signals derived from the generation process.

## 3.2 OVERVIEW OF TRACEDET

**Denoising as a Markov Decision Process**. The major challenge of hallucination detection in D-LLMs is the mismatch between the intermediate model generation and the final response. More specifically, it is challenging to determine how hallucination arises in the final response and utilize this information for detection, as partial information in the generated sequence can be erased during multi-step denoising and remasking. Essentially, the denoising process of D-LLMs can be formulated as a Markov Decision Process (MDP) over decoding steps (Black et al., 2024):

- **State**: We define the $t$-th state $s_t = (p_0, r_{T-t})$ in the denoising process as the combination of the input query $p_0$ and the intermediate sequence $r_{T-t}$ at the $t$-th denoising step.

- **Action**: At the $t$-the state, the D-LLM predicts all masked tokens from $s_t$ and generates a full token sequence $\hat{r}_{T-t-1}$ from distribution $P_\theta(r_0 \mid r_{T-t}, p_0)$.

- **Transition**: After generating $\tilde{r}_{T-t-1}$, a fraction of tokens is remasked to form $r_{T-t-1}$ depending on the noise schedule and remasking strategy. Then, the D-LLM moves to the next state $s_{t+1} = (p_0, r_{T-t-1})$ to start another round of denoising.

**Hallucination Detection from Action Trace**. With the MDP formulation, we can leverage the entire action trace $A = \{a_0, a_1, \ldots, a_{T-1}\}$ rather than only the final response $r_0$ to detect hallucinations. Each action reveals how the model progressively refines its generation. Our observations in Figure 1 show that hallucinated outputs are strongly associated with intermediate denoising steps,

especially when the intermediate outputs contains distracting or ambiguous content. Additional examples are provided in Appendix F. TraceDet exploits this insight by discovering hallucination-relevant action sub-trace $A_{sub}$ from $A$, and then trains a classifier $f$ on $A_{sub}$ to distinguish hallucination from factual responses:

$$\min_{f,g} \mathcal{L}(Y, f(A_{sub})), \quad s.t. \quad A_{sub} = g(A). \tag{2}$$

Here $g(\cdot)$ is a neural network that identifies $A_{sub}$ from $A$. By capturing hallucination signals throughout the entire action trace, TraceDet provides an interpretable and fine-grained perspective on how hallucinations emerge during the denoising trajectory. Moreover, TraceDet can be applied to diverse D-LLMs with different noise schedules and remasking strategies.

### 3.3 OPTIMIZATION OF TRACEDET VIA INFORMATION-THEORETIC APPROACH

A key challenge of TraceDet lies in the fact that hallucination-relevant actions in the denoising trajectory of D-LLMs are not known *a priori*. The hallucination-relevant action may be sparse and unevenly distributed across the action trace, and not every action contributes equally to the emergence of hallucination. This necessitates learning to identify such a sub-trace $A_{sub}$ from $A$ that is informative to the hallucinated response, in the absence of explicitly labeled $A_{sub}$. Inspired by the information bottleneck principle (Tishby et al., 2000; Tishby & Zaslavsky, 2015), TraceDet reformulates the objective in Eq. 2 from an information-theoretic perspective:

$$\min_{\substack{f:A_{\text{sub}} \mapsto Y \\ g:A \mapsto A_{\text{sub}}}} -I(Y; A_{\text{sub}}) + \beta I(A; A_{\text{sub}}), \tag{3}$$

where $I(A; B) = \iint_{A,B} P(A, B) \log \frac{P(A,B)}{P(A)P(B)} \mathrm{d}A \mathrm{d}B$ is the mutual information between the random variable $A$ and $B$. The first term $I(Y; A_{sub})$ in Eq. 3 encourages the identified $A_{sub}$ is relevant to the hallucination and the second term $I(A; A_{sub})$ regularizes the identified $A_{sub}$ only contains partial information of $A$ to avoid the trivial solution $A_{\text{sub}} = A$. $\beta$ is the hyperparameter to trade off between the two terms. By trading off between the two terms in Eq. 3, TraceDet aims to identify the *minimally sufficient* sub-trace in the denoising process of D-LLMs for hallucination detection.

Furthermore, as the mutual information related objectives are often intractable and hard to optimize, we need to resort more practical forms of Eq. 3. We begin by examining the first term in Eq. 3:

$$-I(Y; A_{\text{sub}}) \leq \mathbb{E}_{Y,A_{\text{sub}}}[-\log q_\theta(Y \mid A_{\text{sub}})] := \mathcal{L}_{\text{cls}}(f(A_{\text{sub}}), Y). \tag{4}$$

Here $q_\theta(Y \mid A_{\text{sub}})$ is the variational approximation to $p(Y \mid A_{\text{sub}})$, which corresponds to the classifier $f(\cdot)$ that predicts whether the identified sub-trace is relevant to the hallucination. And $\mathcal{L}_{\text{cls}}(\cdot)$ is the classification loss, and we choose the cross-entropy loss in practice. Then, we proceed to derive the upper bound of the second term $I(A; A_{\text{sub}})$ in Eq. 3:

$$I(A; A_{\text{sub}}) = \mathbb{E}_{A,A_{\text{sub}}} \left[ \log \frac{P(A_{\text{sub}} \mid A)}{Q(A_{\text{sub}})} \right] - D_{\text{KL}}\big(P(A_{\text{sub}}) \,\|\, Q(A_{\text{sub}})\big)$$

$$\leq \mathbb{E}_A \left[ D_{\text{KL}}\big(P(A_{\text{sub}} \mid A) \,\|\, Q(A_{\text{sub}})\big) \right], \tag{5}$$

where $D_{\text{KL}}(\cdot \| \cdot)$ is the KL-divergence(Kullback & Leibler, 1951). The inequality in Eq. 5 is induced using the non-negative nature of the KL-divergence. Recall that $A = \{a_0, a_1, \ldots, a_{T-1}\}$, the posterior distribution $P(A_{sub} \mid A)$ can be factorized into $\Pi_{a_i \in A} \text{Bernoulli}(p_{a_i})$ where we assume that the sub-instance extractions of $A$ are independent. The corresponding prior distribution $Q(A_{sub})$ in Eq. 5 is the non-informative distribution $\Pi_{a_i \in A} \text{Bernoulli}(\tau)$, where $\tau$ restricts the proportion of traces that will be selected. To this end, we relax Eq. 5 and approximate it by the following differentiable objective as discussed in Appendix D:

$$\mathcal{L}_{\text{ext}} = \sum_{i=0}^{T-1} \left[ p_{a_i} \log \frac{p_{a_i}}{\tau} + (1 - p_{a_i}) \log \frac{1 - p_{a_i}}{1 - \tau} \right], \tag{6}$$

where $p_{a_i}$ is the predicted probability to select trace $a_i$. For consistency and unification, we train the composed function $f \circ g(A)$ with the following learning objective:

$$\mathcal{L} = \mathcal{L}_{\text{cls}} + \beta \mathcal{L}_{\text{ext}}, \tag{7}$$

where $\beta$ is the same hyperparameter as in Eq. 3, controlling the strength of the regularization term.

Table 1: AUROC(%) comparison of hallucination detection methods on two D-LLMs across three QA datasets with 128 and 64 generation step lengths. SS is the short for Single Sampling. The highest score is **bolded** and the second highest is underlined.

| Model | Method | SS | TriviaQA 128 | TriviaQA 64 | HotpotQA 128 | HotpotQA 64 | CommonsenseQA 128 | CommonsenseQA 64 | Ave |
|---|---|---|---|---|---|---|---|---|---|
| | | | Output-Based | | | | | | |
| | Perplexity | ✗ | 50.4 | 47.6 | 49.3 | 51.2 | 65.6 | 65.0 | 54.9 |
| | LN-Entropy | ✗ | 54.6 | 53.5 | 54.8 | 54.7 | 64.6 | 64.4 | 57.8 |
| | Semantic Entropy | ✗ | 68.9 | 67.3 | 57.6 | 53.8 | 44.1 | 43.9 | 55.9 |
| | Lexical Similarity | ✗ | 62.5 | 59.0 | 64.2 | 57.1 | 57.3 | 60.7 | 60.1 |
| LLaDA-8B-Instruct | | | Latent-Based | | | | | | |
| | EigenScore | ✗ | 69.2 | 66.9 | 64.7 | 59.2 | 58.5 | 60.6 | 63.2 |
| | CCS | ✓ | 57.1 | 54.2 | 57.6 | 55.8 | 50.5 | 58.5 | 55.6 |
| | TSV | ✓ | 60.2 | 61.1 | 65.0 | 59.4 | 52.9 | 55.2 | 59.0 |
| | TraceDet | ✓ | **73.9** | **74.1** | **66.1** | **63.7** | **77.2** | **77.1** | **72.0** |
| | | | Output-Based | | | | | | |
| | Perplexity | ✗ | - | - | - | - | - | - | - |
| | LN-Entropy | ✗ | - | - | - | - | - | - | - |
| | Semantic Entropy | ✗ | 73.7 | 72.5 | 62.7 | 67.7 | 51.4 | 48.6 | 62.8 |
| | Lexical Similarity | ✗ | 58.3 | 64.0 | 59.7 | 62.7 | 77.3 | 76.9 | 66.5 |
| Dream-7B-Instruct | | | Latent-Based | | | | | | |
| | EigenScore | ✗ | 66.0 | 69.1 | 62.5 | 67.0 | 76.9 | 77.5 | 69.8 |
| | CCS | ✓ | 56.9 | 50.3 | 51.7 | 58.2 | 54.2 | 53.2 | 54.1 |
| | TSV | ✓ | 75.6 | 74.7 | 58.7 | 63.0 | 62.3 | 56.8 | 65.2 |
| | TraceDet | ✓ | **78.1** | **86.7** | **75.1** | **76.0** | **84.7** | **84.1** | **80.8** |

## 3.4 IMPLEMENTATION

For our method, we define the action trace using distributional statistics, specifically the token-wise entropy trace derived from $P_\theta$. This choice captures the temporal evolution of uncertainty during generation while yielding a fixed-size representation. Alternatively, one could construct action traces from the token embeddings of $\tilde{r}$. However, embedding-based traces, particularly when combined with temporal encodings, make the representation extremely large and often introduce severe numerical instabilities in practice. Our detection model is decomposed into two learnable modules: the **sub-instance extractor** $g_\theta$ and the **sub-instance predictor** $f_\phi$, with implementation and complexity details discussed in Appendix I:

(a) **Sub-instance extractor**: Given the entropy sequence $A \in \mathbb{R}^{T \times B \times D}$, we concatenate it with sinusoidal time embeddings and encode the result using a Transformer (Queen et al., 2023; Liu et al., 2024), yielding contextual embeddings $emb$. The extractor then produces a probabilistic mask $\hat{M} \in (0, 1)^{T \times B}$, where $\hat{m}_{t,b} = \text{Linear}(\text{attn}(emb, A))$, with attn a cross-attention mechanism using $emb$ as query and a representation of $A$ as key/value. A temporal binary mask $M \in \{0, 1\}^{T \times B}$ is then sampled from $\hat{M}$ and applied to $A$, i.e., $A_{\text{sub}} = M \odot A \in \mathbb{R}^{T \times B \times D}$, where $\odot$ denotes element-wise multiplication. However, the mask sampling process is inherently non-differentiable, so we employ the Gumbel–Softmax trick (Jang et al., 2016).

(b) **Sub-instance predictor**: The masked trajectory $A_{\text{sub}}$ is temporally aggregated and passed to the predictor $f_\phi$, which directly outputs hallucination probabilities $f_\phi(A_{\text{sub}}) \in [0, 1]$, $b = 1, \ldots, B$. In practice, $f_\phi$ consists of temporal aggregation followed by an MLP and an activation layer.

## 4 EXPERIMENTS

### 4.1 SETUP

**Datasets**. We conduct experiments on three widely used factuality QA benchmarks: **TriviaQA** (Joshi et al., 2017) consists of open-domain factoid questions with answer spans in Wikipedia. **CommonsenseQA** (Talmor et al., 2018) contains multiple-choice questions requiring commonsense reasoning. **HotpotQA** (Yang et al., 2018) contains multi-hop questions requiring aggregation across

Table 2: F1 score (%) comparison between TraceDet and baseline methods. The highest score is **bolded**.

| Method | TriviaQA | | HotpotQA | | CommonsenseQA | |
|---|---|---|---|---|---|---|
| | 128 | 64 | 128 | 64 | 128 | 64 |
| Lexical Similarity | 49.9 | 50.6 | 59.8 | 55.2 | 80.0 | 82.6 |
| EigenScore | 56.0 | 51.8 | 63.9 | 56.9 | 79.8 | 82.9 |
| CCS | 59.3 | 65.0 | 47.3 | 53.0 | 61.0 | 53.5 |
| TSV | 70.5 | 67.6 | 68.5 | 65.1 | 62.5 | 57.6 |
| TraceDet | **76.7** | **80.2** | **73.8** | **68.7** | **89.7** | **90.2** |

Table 3: AUROC(%) comparison between TraceDet and our proposed baselines (Ave Entropy, TraceDet w/o Masking). The highest score is **bolded** and the second highest is underlined.

| Model | Method | TriviaQA | | HotpotQA | | CommonsenseQA | | Ave |
|---|---|---|---|---|---|---|---|---|
| | | 128 | 64 | 128 | 64 | 128 | 64 | |
| **LLaDA-8B-Instruct** | Ave Entropy | 61.3 | 68.3 | 56.2 | 58.1 | 63.8 | 68.8 | 62.8 |
| | TraceDet w/o Masking | 71.2 | 70.3 | 63.2 | 61.6 | 73.1 | 75.2 | 69.1 |
| | TraceDet | **73.9** | **74.1** | **66.1** | **63.7** | **77.2** | **77.1** | **72.0** |
| **Dream-7B-Instruct** | Ave Entropy | 59.1 | 69.9 | 50.8 | 57.8 | 77.1 | 76.9 | 65.3 |
| | TraceDet w/o Masking | 76.2 | **87.1** | 72.5 | 74.1 | 81.4 | 79.1 | 78.4 |
| | TraceDet | **78.1** | 86.7 | **75.1** | **76.0** | **84.7** | **84.1** | **80.8** |

supporting contexts. These datasets enable the evaluation of hallucination detection across varying reasoning complexity. For each dataset, we randomly sample 400 QA pairs from the validation split with available ground-truth labels, partitioned into 200 validation and 200 testing instances to ensure computational efficiency while maintaining task coverage.

**Baseline Methods**. We compare our method against seven baselines spanning two categories proposed in Section 2: (1) Output-based methods: **Perplexity** (Ren et al., 2022), Length-Normalized Entropy (**LN-Entropy**) (Malinin & Gales, 2020), **Semantic Entropy** (Kuhn et al., 2023), and **Lexical Similarity** (Lin et al., 2023); (2) Latent-based methods: **EigenScore** (Chen et al., 2024a), Contrast-Consistent Search (**CCS**) (Burns et al., 2022), and Truthfulness Separator Vector (**TSV**) (Park et al., 2025). We also include two TraceDet variants. **Ave Entropy** uses the average of stepwise entropies as a naive confidence measure. In **TraceDet w/o Masking**, we train a transformer detector while removing the sub-step extraction and its associated loss. All methods use identical dataset splits, random seeds, and configurations to ensure fair comparison.

**Models**. We adopt **Dream-7B-Instruct** (Ye et al., 2025) and **LLaDA-8B-Instruct** (Nie et al., 2025) as representative D-LLMs in this work. To the best of our knowledge, they are the only opensource D-LLMs that provide stepwise token-level logits and hidden representations necessary for comprehensive baseline comparison across both Output-based and Latent-based detection methods.

**Evaluation**. We employ task-specific evaluation protocols: multiple-choice tasks are evaluated by direct comparison with the ground-truth, while Qwen3-8B (Yang et al., 2025a) serves as an external judge for hallucination assessment in open-domain QA. We further measured the agreement between the Qwen3-8B judge and human evaluation, finding 90% consistency on TriviaQA and 84% on HotpotQA. Following previous work (Park et al., 2025; Chen et al., 2024a), we report AUROC scores, with model selection based on validation performance and evaluation on held-out test sets.

## 4.2 MAIN RESULTS

Table 1 presents a comprehensive comparison of TraceDet against baseline hallucination detection methods across two D-LLMs and three factuality QA datasets with varying generation lengths. TraceDet achieves the highest performance in all experimental settings, outperforming the second strongest baseline by 8.8% AUROC on LLaDA-8B-Instruct and 11% on Dream-7B-Instruct. These consistent gains exhibit the value of exploiting temporal denoising dynamics rather than relying solely on output uncertainty or static hidden-state representations. Among the baselines, Outputbased approaches suffer from poor consistency, with LN-Entropy and Perplexity methods showing high variance across datasets on the LLaDA-8B-Instruct model. Due to restricted access to inter-

Table 4: Average inference time of 100 samples for different methods.

| Metric | Perplexity | LN-Entropy | Semantic Entropy | Lexical Similarity | EigenScore | CCS | TSV | TraceDet |
|---|---|---|---|---|---|---|---|---|
| Time (s) ↓ | 468.44 | 710.63 | 801.35 | 715.35 | 693.46 | 140.73 | 160.31 | **147.52** |

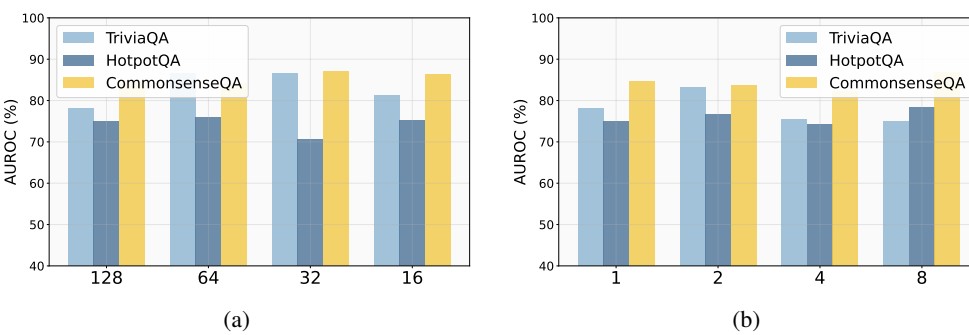

(a)                                      (b)

Figure 3: (a) TraceDet performance of different generation lengths with step length fixed at 1. (b) TraceDet performance with different step lengths with generation length fixed at 128. All results are reported as AUROC using Dream-7B-Instruct.

mediate logits from Dream-7B-Instruct, our attempts to estimate Perplexity and LN-Entropy suffer from severe numerical instabilities, often diverging to infinity. Moreover, Semantic Entropy achieves $75.1\%$ AUROC on TriviaQA with Dream-7B-Instruct but collapses to $51.4\%$ on CommonsenseQA, underscoring poor robustness when hallucinations decouple from token-level ambiguity. Latent-based methods capture hidden-state geometry and show more competitive results, but they remain sensitive to dataset and model choice. TSV achieves $75.6\%$ AUROC on TriviaQA with Dream-7B-Instruct yet fluctuates significantly across tasks while consistently underperforming TraceDet. This superiority is not limited to AUROC, TraceDet also achieves the highest F1 scores across all dataset–model combinations, further confirming its robustness illustrated in Table 2.

To isolate the contributions of our training framework and the IB principle, we evaluate two ablations in Table 3: Ave Entropy and TraceDet w/o Masking. Naively averaging stepwise entropy yields insufficient performance, while training without masking improves results but still underperforms TraceDet. This demonstrates that the IB principle enables TraceDet to identify maximally informative sub-instances, yielding clear improvements over both simplified variants.

## 4.3 ANALYSIS

**Efficiency**. TraceDet is highly efficient at inference. Unlike AR-LLMs, D-LLMs generate text through iterative denoising, where each forward pass is computationally more costly. This makes inference efficiency especially important for hallucination detection in D-LLMs. Existing methods impose significant computational overhead through two types of multi-sample computations: (i) Monte-Carlo sampling over output log-likelihood for estimating Perplexity or LN-Entropy, requiring typically at least 128 remasking samples for stable results (Nie et al., 2025), and (ii) response sampling for similarity-based metrics like Lexical Similarity or Semantic Entropy, multiplying inference cost by the number of samples $S$. TraceDet eliminates this overhead by directly leveraging stepwise entropy signals naturally exposed during denoising, requiring no additional sampling. Table 4 demonstrates that TraceDet achieves better performance while significantly reducing inference cost compared to multi-sampling baselines. All timing results are reported on the LLaDA-8B-Instruct model on the same hardware device with $S$ set to 10, which aligns with practical application settings.

**Sensitivity to Generation Length and Step Length**. D-LLMs typically generate sequences with fixed lengths, making parameter sensitivity analysis crucial for practical deployment. Generation length determines the entropy matrix dimensionality, while step length controls token retention at each denoising step. To assess the influence of these parameters, we conduct sensitivity analysis by varying generation length $L \in \{16, 32, 64, 128\}$ with step length fixed at 1 (Figure 3a), and step length $S \in \{1, 2, 4, 8\}$ with generation length fixed at 128 (Figure 3b).

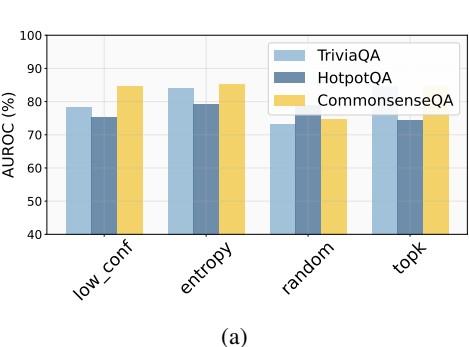 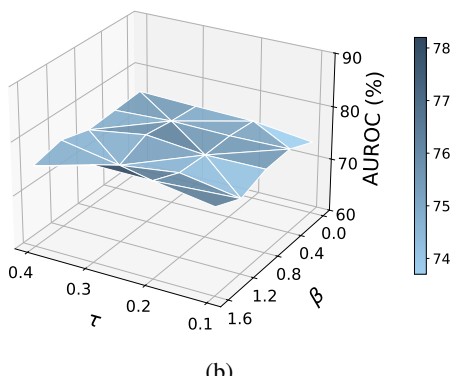

(a)        (b)

Figure 4: (a) TraceDet performance sensitivity to remasking strategies. (b) TraceDet performance sensitivity to $\mathcal{L}_{ext}$ parameters $\tau$ and $\beta$ on TriviaQA. All results are reported as AUROC using Dream-7B-Instruct.

TraceDet demonstrates robust performance across parameter ranges. As shown in Figure 3a, TraceDet achieves optimal performance at moderate lengths (64 and 32 tokens), with slight deterioration at the longest setting (128 tokens). This suggests that excessively long sequences may introduce noise that dilutes the hallucination signal. For fact-based QA tasks like TriviaQA, where answers are typically concise, generation length 64 provides sufficient reasoning capacity while maintaining detection quality. Figure 3b shows that step length has minimal impact on performance, with all settings yielding comparable results across datasets. The consistent performance across both parameter dimensions indicates that TraceDet's effectiveness is not critically dependent on generation settings, making it practically robust for deployment across diverse D-LLM configurations.

**Sensitivity to Remasking Strategies**. Figure 4a examines the impact of different remasking strategies on TraceDet's performance with Dream-7B-Instruct across four approaches: low-confidence (retaining most confident predictions), entropy (retaining lowest entropy tokens), random (random retention), and top-k (retaining based on top-1/top-2 confidence margins). TraceDet maintains robust performance across most strategies, with AUROC scores ranging from 75-85% on TriviaQA and CommonsenseQA. However, random remasking shows degraded performance on TriviaQA, likely because random token retention disrupts the model's ability to maintain coherent reasoning patterns essential for fact-based question answering. The stability across remasking strategies (low-confidence, entropy, top-k) demonstrates TraceDet's adaptability to different D-LLM configurations.

**Sensitivity to Hyperparameters**. Figure 4b analyzes the sensitivity of TraceDet to the $\mathcal{L}_{ext}$ hyperparameters $\tau$ (masking ratio) and $\beta$ (regularization weight). The 3D surface plot reveals a stable performance plateau across a wide range of parameter combinations. TraceDet achieves optimal performance when $\tau \in [0.2, 0.3]$ and $\beta \in [0.8, 1.6]$, indicating that retaining 20-30% of denoising steps with moderate regularization provides the best balance between information preservation and noise reduction. The broad stability region demonstrates that TraceDet does not require precise hyperparameter tuning for effective deployment.

## 5 CONCLUSION

In this work, we addressed the challenge of hallucination detection in D-LLMs by introducing a new framework, **Decoding Trace Detection (TraceDet)**. TraceDet is a lightweight, diffusion model architecture–aware detector built upon information bottleneck principles, which identifies sufficient sub-instances from the denoising entropy matrix. Our experiments demonstrate that **TraceDet** consistently achieves superior performance on mainstream D-LLMs across multiple datasets. Beyond proposing a new hallucination detection method, this work also offers insights into the mechanisms of hallucination generation in D-LLMs, paving the way toward building more reliable applications of D-LLMs. As future work, we will focus on developing strategies to mitigate the proposed hallucinated patterns. We believe the insights from TRACEDET can inspire future methods that more effectively leverage decoding traces as reliable supervision signals for improved detection.

ACKNOWLEDGMENT

The authors sincerely thank the SAC, AC, and all the reviewers for their valuable time and insightful feedback, which have significantly improved the quality of this work. This work is supported by the UKRI grant: Turing AI Fellowship EP/W002981/1. The authors also thank Hertford College, University of Oxford, for supporting this research through its Summer Research Studentships programme.

ETHICS STATEMENT

This research did not involve identifiable human data or animals and therefore did not require approval from an institutional ethics committee or review board. All experiments are conducted on publicly available datasets for scientific purposes only. The work does not involve or target any sensitive attributes such as gender, race, nationality, or skin color. Our study focuses on hallucination detection in diffusion large language models, with the aim of improving the reliability and safety of LLMs.

REPRODUCIBILITY STATEMENT

We have made every effort to ensure the reproducibility of our work. All experiments were conducted using publicly available datasets, and we provide detailed descriptions of data preprocessing in Section 4. Our model architecture, hyperparameters, and training protocols are fully specified in Section 3 and Appendix E. We will release our code and scripts for data processing and evaluation upon publication to facilitate replication.

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

## A    USAGE OF LLMS

For transparency, we report our use of LLMs. We employed OpenAI ChatGPT solely to assist with language polishing, copy editing, and improving exposition in terms of grammar, phrasing, and organization. In addition, we used Qwen-8B as an external judge for hallucination detection in open-domain QA, but its outputs were carefully checked by the authors. None of these LLMs was used to (i) generate scientific hypotheses or claims; (ii) design experiments; (iii) perform data collection, processing, or analysis; (iv) compute metrics or produce numerical results; or (v) create figures. All substantive content, experimental procedures, and numerical results were produced by the authors. The corresponding author assumes full responsibility for ensuring the accuracy and integrity of the paper.

## B    DETAILED RELATED WORK

**Baseline Methods**. We compare our method against seven baselines spanning two categories proposed in Section 4.

*(1) Output-based methods*: These methods operate only on the generated text, detecting hallucinations by analyzing uncertainty or surface similarity.

- **Perplexity** (Ren et al., 2022): computes the negative log-likelihood of the generated sequence under the base model. Higher perplexity indicates that the model assigns low probability to its own output, suggesting potential hallucination.

- **Length-Normalized Entropy (LN-Entropy)** (Malinin & Gales, 2020): measures the token-level predictive entropy of the output distribution, normalized by sequence length, so that generations with unusually high average uncertainty are flagged as hallucinations.

- **Semantic Entropy** (Kuhn et al., 2023): measures consistency of multiple generations by partitioning them into semantic classes and computing the entropy of this class distribution. A higher semantic entropy indicates greater uncertainty, which is taken as a signal of hallucination.

- **Lexical Similarity** (Lin et al., 2023): assesses consistency of multiple generations using lexical overlap metrics. Low overlap suggests divergence from supporting evidence, which is interpreted as hallucination.

*(2) Latent-based methods.* These approaches exploit hidden states or latent directions of LLMs, probing truthfulness signals directly from internal representations.

- **EigenScore** (Chen et al., 2024a): proposed in INSIDE (ICLR 2024), it measures response consistency via the log-determinant of the covariance matrix of their latent embeddings.

- **Contrast-Consistent Search (CCS)** (Burns et al., 2022): queries the model with contrastive prompts (e.g., factual vs. hallucinated) and trains a MLP to evaluate consistency of latent representations. Inconsistent activations are interpreted as evidence of hallucination.

- **Truthfulness Separator Vector (TSV)** (Park et al., 2025): Trains a steering vector that separates truthful from hallucinated generations in latent space, and then classifies new samples by projecting onto the learned centroids.

**Other potential baselines**. Several alternative baselines for hallucination detection exist, however they are challenging to implement in comparison to our proposed method. One major category, highlighted in Huang et al. (2025), is fact-checking external retrieval methods, including: **FactScore** (Min et al., 2023), which decomposes long-form text into fact chunks and computes the proportion of chunks verified by an external knowledge base. And, **Factool** (Chern et al., 2023), a tool-augmented method enabling LLMs to detect factual hallucinations using external resources. These approaches rely on additional knowledge sources (e.g., Wikipedia or local databases), which conflicts with our core objective of detecting hallucinations *without external verification*.

Other recent hallucination methods include:

- **ReDeEP** (Sun et al., 2024), a method specifically designed for Retrieval-Augmented Generation (RAG). It disentangles retrieved evidence from the LLM's parametric knowledge, then measures

the alignment between the two. While effective in RAG settings, it is task-specific: in our internal-signal-based experiments, there is no retrieved context to disentangle, so ReDeEP is not applicable.

- **FactTest** (Nie et al., 2024), which formulates hallucination detection as a distribution-free hypothesis testing problem. It controls Type I error by introducing an abstention mechanism, requiring a held-out calibration dataset and repeated testing procedures. Although statistically elegant, this framework is fundamentally different from our design goal: we aim for lightweight detection relying purely on model-internal dynamics, without the need for extra calibration data or abstention strategies.

- **AGSER** (Liu et al., 2025a), which leverages attention-guided self-reflection. It analyzes self-attention maps, distinguishes "attentive" vs. "non-attentive" tokens, and trains a secondary classifier on the distribution of attention patterns to identify hallucinations. This approach depends on full access to intermediate attention weights and assumes an autoregressive token-to-token attention structure. However, D-LLMs adopt denoising architectures where attention is not aligned with autoregressive decoding, making AGSER difficult to adapt. Additionally, extracting and processing large attention tensors incurs substantial computational overhead, which goes against our efficiency-oriented design.

Overall, while these methods are valuable in their respective contexts, their reliance on external resources, additional datasets, or architectural assumptions makes them unsuitable as direct baselines for our study. For this reason, we do not include them in the main comparison and instead discuss them here for completeness.

## C  ALGORITHMS

We provide the pseudo-code of our method in Algorithm 1.

## D  MORE ON EQ 4 AND EQ 5

### D.1  EQ 4

$$
\begin{aligned}
I(Y; A_{\text{sub}}) &= \int P(y, A_{\text{sub}}) \log \frac{P(y \mid A_{\text{sub}})}{P(y)} \, dy \, dA_{\text{sub}} \\
&= \int P(y, A_{\text{sub}}) \log P(y \mid A_{\text{sub}}) \, dy \, dA_{\text{sub}} - \int P(y, A_{\text{sub}}) \log P(y) \, dy \, dA_{\text{sub}} \\
&= \int P(y, A_{\text{sub}}) \log P(y \mid A_{\text{sub}}) \, dy \, dA_{\text{sub}} + H(Y) \\
&\geq \int P(y, A_{\text{sub}}) \log q_\theta(y \mid A_{\text{sub}}) \, dy \, dA_{\text{sub}} \\
&= \mathbb{E}_{Y, A_{\text{sub}}} \big[ \log q_\theta(Y \mid A_{\text{sub}}) \big] := -\mathcal{L}_{cls}
\end{aligned}
$$

---

**Algorithm 1** Overall training framework for TRACEDET

1: **Parameters:** batch size $B$, prior masking ratio $r$, extraction weight $\beta$
2: **Inputs:** Entropy sequence $A \in \mathbb{R}^{T \times B \times D}$
3: **Initialize:** Transformer encoder with random weights
4: **Input encoding:**
5: Project $A$ using MLP-based positional encoder
6: Concatenate sinusoidal time embeddings of dimension $d_{\text{pe}}$
7: Apply Transformer encoder to obtain contextual embeddings $\text{emb} \in \mathbb{R}^{T \times B \times d_{\text{ff}}}$
8: **Sub-instance extraction:**
9: **for** $t = 1$ to $T$ **do**
10:      **for** $b = 1$ to $B$ **do**
11:          Compute cross attention: $att = \text{attn}(\text{emb}, A)$
12:          Apply linear layer and softmax over time steps: $\hat{m}_{t,b} = \text{softmax}(\text{Linear}(att))$
13:      **end for**
14: **end for**
15: Apply masking: $A_{\text{sub}} = M \odot A$
16: **Sub-instance classification:**
17: **for** $b = 1$ to $B$ **do**
18:      Aggregate temporally: $A'_{\text{sub}} = \text{Mean}(A_{\text{sub},\cdot,b,\cdot})$
19:      Compute hallucination probability: $\hat{y}_b = \text{ReLU-MLP}(A'_{\text{sub}})$
20: **end for**
21: **Training objective:**
22: Classification loss: $\mathcal{L}_{\text{cls}} = \text{BCE}(\hat{y}_b, y_b)$
23: Extraction regularizer:

$$\mathcal{L}_{\text{ext}} = \sum_{t=1}^{T} \sum_{b=1}^{B} \left[ \hat{m}_{t,b} \log \frac{\hat{m}_{t,b}}{r} + (1 - \hat{m}_{t,b}) \log \frac{1 - \hat{m}_{t,b}}{1 - r} \right]$$

24: Total loss: $\mathcal{L} = \mathcal{L}_{\text{cls}} + \beta \, \mathcal{L}_{\text{ext}}$
25: Backpropagate and update parameters

---

## D.2 EQ 5

$$
\begin{aligned}
I(A; A_{\text{sub}}) &= \mathbb{E}_{A,A_{sub}} \left[ \log \frac{P(A_{sub} \mid A)}{P(A_{sub})} \right] \\
&= \mathbb{E}_{A,A_{sub}} \left[ \log \frac{P(A_{sub} \mid A) Q(A_{sub})}{P(A_{sub}) Q(A_{sub})} \right] \\
&= \mathbb{E}_{A,A_{sub}} \left[ \log \frac{P(A_{sub} \mid A)}{Q(A_{sub})} \right] - \mathbb{E}_{A,A_{sub}} \left[ \log \frac{Q(A_{sub})}{P(A_{sub})} \right] \\
&= \mathbb{E}_{A,A_{sub}} \left[ \log \frac{P(A_{sub} \mid A)}{Q(A_{sub})} \right] - D_{\text{KL}}(P(A_{sub}) \| Q(A_{sub})) \\
&\leq \mathbb{E}_{A} \left[ D_{\text{KL}}\big(P(A_{\text{sub}} \mid A) \, \| \, Q(A_{\text{sub}})\big) \right]
\end{aligned}
$$

## D.3 PROOF OF RELAXATION

**KL decomposition under independent Bernoulli factors.** Under the independence assumption, both the true posterior $P(A_{\text{sub}} \mid A)$ and the variational prior $Q(A_{\text{sub}})$ factorize into products of Bernoulli distributions. In this case, the KL divergence decomposes into a sum of element-wise Bernoulli KL terms:

$$D_{\text{KL}}\big(P(A_{\text{sub}} \mid A) \, \big\| \, Q(A_{\text{sub}})\big) = \sum_{i} D_{\text{KL}}\big(\text{Bern}(p_{a_i}) \, \big\| \, \text{Bern}(\tau)\big).$$

**KL divergence between two Bernoulli variables.** For a single Bernoulli distribution, the KL divergence has the following closed form:

$$D_{\mathrm{KL}}\big(\mathrm{Bern}(p)\,\big\|\,\mathrm{Bern}(\tau)\big) = p\log\frac{p}{\tau} + (1-p)\log\frac{1-p}{1-\tau}.$$

## E  Experiment Settings

All experiments were conducted on NVIDIA A40 GPUs. Hyperparameter settings are shown in Tabel 5.

Table 5: Hyperparameter search space for TRACEDET. *Notation:* [†] log-spaced; [‡] linearly spaced. [*] only applies to LLaDA.

| Parameter | Range | Grid size |
|---|---|---|
| Learning rate ($lr$) | $[10^{-5},\,10^{-3}]^{\dagger}$ | 8 |
| Batch size ($batch\_size$) | $\{8,\,64\}$ | 2 |
| Dropout rate ($dropout\_rate$) | $[0.0,\,0.4]^{\ddagger}$ | 5 |
| Number of layers ($nlayers$) | $\{2,\,3,\,4\}$ | 3 |
| $\beta$ | $[0,\,2]^{\ddagger}$ | 6 |
| $\tau$ | $[0.1,\,0.4]^{\ddagger}$ | 4 |
| cfg[*] | $\{0,\,1\}$ | 2 |

## F  Case Study

### F.1  Interleaving Hallucination

---

**Retained Steps**

<Question>Professor A. Selvanathan is a professor at a university that is public or private?</Question>
<Golden label>public</Golden label>
The following are TraceDet extracted steps
<step>"<answer>public</answer>", </step>
<step>"<answer>Public</answer>", </step>
<step>"<answer>private</answer>", </step>
<step>"<answer>private</answer>", </step>
<Output>"<answer>private</answer>"</Output>

---

**Retained Step**

<Question>Friggatriskaidekaphobia (or triskaidekaphobia or paraskevidekatriaphobia) is the fear of what?</Question>
<Golden label>Friday the 13</Golden label>
The following are TraceDet extracted steps
<step>"<answer> Friggriskagk anythingobia (or frost excessive or kevide atr13 is the fear of numbers.</answer>",</step>
<step>"<answer> Friggriskak thingsobia (oriska k refrigeration oranswerkevide atriobia) is the fear of ice. </answer>", </step>
<step>"<answer>Friggatriskakobia (oriskak orkevideatri) is fear of freezing.</answer>", </step>
<Output>"<answer>Friggatriskaidekaphobia (or triskaidekaphobia or paraskevidekatriaphobia) is the fear of numbers.</answer>"</Output>

---

> **Retained Steps**
>
> <Question>On which of the hills of ancient Rome were the main residences of the Caesars?</Question>
> <Golden label>Palatine</Golden label>
> The following are TraceDet extracted steps
> <step>"<answer>Palatine</answer>", </step>
> <step>"<answer>Palatine</answer>", </step>
> <step>"<answer>Pal Hill Hill</answer>", </step>
> <step>"<answer>Palat Hill</answer>", </step>
> <Output>"<answer>Palat Hill</answer>"</Output>

> **Retained Steps**
>
> <Question>What NIFL Premier Intermediate League team did Sean Connor play for?</Question>
> <Golden label>Distillery</Golden label>
> The following are TraceDet extracted steps
> <step>"<answer>Distillery F.C</answer>",</step>
> <step>"<answer>Newis Distillery F.C</answer>", </step>
> <step>"<answer>Newington Youth.C.C.</answer>", </step>
> <step>"<answer>Newington Youth F.C.</answer>", </step>
> <Output>"<answer>Newington Youth F.C.</answer>"</Output>

## F.2 INCONSISTENT GUESSES

> **Retained Step**
>
> <Question>Who was declared Model of the Millennium by Vogue editor Anna Wintour?</Question>
> <Golden label>Gisele Buendchen</Golden label>
> The following are TraceDet extracted steps
> <step>"<answer> Cindy Campbell Crawford declared declared Millennium Millennium editor editor editor editor editor editorour editor editor editor editor editor editorlags Campbell</answer>",</step>
> <step>"<answer> Cindy Crawford Crawford declared declared declared declared Millennium Millennium editor editor editor editor editorint editorintint Campbell Campbell Campbell Campbelllags Campbell </answer>", </step>
> <step>"<answer>Naomi Crawford Crawford Crawford Mossourour Campbell Campbell Campbell Campbell Campbell Campbell Campbelllags Crawford</answer>", </step>
> <step>"<answer>Nicole Kid Crawford ¡/answer¿lags Campbell Campbell Campbell Nicole Campbell0 model</answer>", </step>
> <Output>"<answer>Nicole Kidman</answer>"</Output>

> **Retained Step**
>
> <Question>Which Canadian born actress was the star in the movie Barb Wire?</Question>
> <Golden label>Pamela Anderson</Golden label>
> The following are TraceDet extracted steps
> <step>"<answer> answeransweransweransweranswer actress actressanswer actress actressanswer actress actress actress actress actress actress actress Wire Wire Wire Wire Wire Wire Wire Wire Wire Wire Barb Wire Wire Wire Wire Wire Wire</answer>",</step>
> <step>"<answer> JenniferJennifer ̆739b ̆7d22 ̆739b ̆7d22 ̆739b ̆7d22answeransweransweranswer Wireanswer Wire Wire Wire Wire Wire</answer>", </step>
> <step>"<answer>Michelleodie Ther ̆739b ̆7d22 ̆739b ̆7d22 ̆55106 ̆069¿ Wire Wire Wire Wire Wire Canadian</answer>", </step>
> <step>"<answer>Kirstodie Therellar ̆55106 ̆069answer¿ anisotropicwald Kirst Kirst Kirstaghanaghan Barb</answer>", </step>
> <Output>"<answer>Kirstie Alley</answer>"</Output>

### F.3 PERSISTENT ERROR

> **Retained Step**
>
> <Question>In the Shakespeare play The Tempest, Prospero is the overthrown Duke of where?</Question>
> <Golden label>The weather in Milan</Golden label>
> The following are TraceDet extracted steps
> <step>" InInIn Shakespeare play Temp Temp Temp Prosper Prosper Prosper Prosper Prosper Prosper Prosperrownrown Duke Duke Prosperiel. Prosper¡¡answer¿In Prosper Shakespeare playThe Tempest, Prospero Prosper Prosper overthrown Duke of 'Ari¿",</step>
> <step>" In the Shakespeare play,o is thethrown Duke of ' overrown Duke of Ž2018ieliel. ¡answer¿ In Shakespeare play Theest,o is the overrown Duke of Ariiel", </step>
> <step>"In Shakespeare play The, is thethrown Duke of 'Ariiel'. In the Shakespeare play Theest, Prospero is the overthrown Duke of 'Ariiel'.", </step>
> <step>"In the Shakespeare play The Tempest, Prospero is the overthrown Duke of 'Ariiel'", </step>
> <Output>"In the Shakespeare play The Tempest, Prospero is the overthrown Duke of 'Ariiel'"</Output>

## G ADDITIONAL QUANTITATIVE RESULTS

Table 6: TPR@FPR=0.1 (%) comparison between TraceDet and baseline methods. The highest score is **bolded**.

| Method | TriviaQA | | HotpotQA | | CommonsenseQA | |
|---|---|---|---|---|---|---|
| | 128 | 64 | 128 | 64 | 128 | 64 |
| Lexical Similarity | 14.5 | 15.6 | 19.8 | 19.3 | 0.0 | 0.0 |
| EigenScore | 23.9 | 29.6 | **23.5** | 24.1 | 40.2 | 26.8 |
| CCS | 9.0 | 26.9 | 11.8 | 10.3 | 14.2 | 19.5 |
| TSV | 23.5 | 26.8 | 14.1 | 16.1 | 23.6 | 14.6 |
| TraceDet | **41.4** | **59.6** | 19.3 | **39.1** | **61.5** | **61.6** |

Table 7: Comparison of average epoch training time across TraceDet and training-based baseline methods on TriviaQA using 1700 training samples.

| Method | Training Time (s) |
|---|---|
| CCS | 3.64 |
| TSV | 19.2 |
| TraceDet | 2.25 |

## H AVERAGED TRACE ENTROPY COMPARISON

The core idea of TraceDet is to select a sub-trace $A_{sub}$ that preserves steps predictive of hallucination. To analyze the statistical properties of the extracted sub-traces, we utilize the stepwise maximum token entropy $H_t^{max}$, and summarize each example by the mean over the selected steps $\bar{E} = \frac{1}{|S|} \sum_{t \in S} H_t^{max}$. We compare three variants: (i) No Masking, which uses the full trajectory ($A_{sub} = A$); (ii) TraceDet w/o $\mathcal{L}_{ext}$, the same architecture trained without the extraction loss; and (iii) TraceDet. Figure 5a shows that TraceDet reduces both the mean and variance of $\bar{E}$ while preserving separation between hallucinated and non-hallucinated examples, indicating that masking serves as an effective regularization mechanism. It reduces the maximum entropy among the selected steps, removing noisy or unstable transitions, while still preserving the entropy contrast that distinguishes hallucinated from non-hallucinated examples. Removing masking weakens this effect. Varying the masking ratio $\tau$ (Fig. 5b) confirms that stronger masking more aggressively removes steps with maximum token entropy without reducing the discriminative separation required for reliable detection.

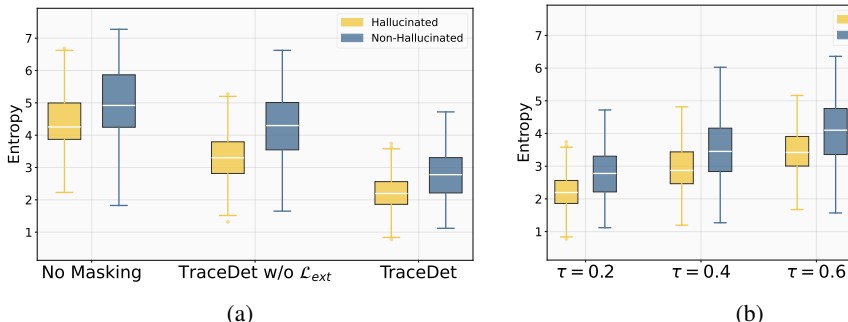

Figure 5: Comparison of averaged trace entropy selected by different model variants. **No masking** refers to the full time step traces. (a) Comparison between different model variants. (b) Comparison between different masking ratios $\tau$. Results are reported using Dream-7B-Instruct.

## I  TRACEDET IMPLEMENTATION DETAILS

- **Sub-instance extractor.** Given the entropy sequence $A \in \mathbb{R}^{T \times B \times D}$, we first concatenate it with sinusoidal time embeddings and encode the result using a lightweight Transformer (2–5 layers, 1 head, feedforward dimension 8), producing contextual embeddings $\mathrm{emb}$. The extractor then generates a probabilistic mask $\hat{M} \in (0,1)^{T \times B}$, where each entry is computed as

$$\hat{m}_{t,b} = \mathrm{Linear}(\mathrm{attn}(\mathrm{emb}, A)),$$

  and $\mathrm{attn}$ denotes a cross-attention module that uses $\mathrm{emb}$ as the query and a learned projection of $A$ as the key/value.

  To obtain a binary temporal mask $M \in \{0,1\}^{T \times B}$, we apply a small Transformer encoder (1 layer, 1 head, feedforward dimension 8) to the concatenation of $A$ and $\mathrm{emb}$, followed by a linear layer and a sampling step. The mask is applied element-wise to $A$ to obtain $A_{\mathrm{sub}} = M \odot A \in \mathbb{R}^{T \times B \times D}$. Since the sampling operation is non-differentiable, we employ the Gumbel–Softmax relaxation to enable gradient-based optimization.

- **Sub-instance predictor.** The masked trajectory $A_{\mathrm{sub}}$ is temporally aggregated and passed to the predictor $f_\phi$, which outputs a hallucination probability for each instance:

$$f_\phi(A_{\mathrm{sub}}) \in [0,1], \qquad b = 1, \ldots, B.$$

In practice, $f_\phi$ consists of a temporal aggregation module followed by a two-layer MLP.

