# OpenReview forum: "TRACEDET: HALLUCINATION DETECTION FROM THE DECODING TRACE OF DIFFUSION LARGE LANGUAGE MODELS"
_ICLR.cc/2026/Conference — ICLR 2026 Poster_

### Official Review · Reviewer_741H · 2025-10-29

**Soundness:** 3
**Presentation:** 3
**Contribution:** 2
**Rating:** 6
**Confidence:** 4

**Summary:**

1) This paper proposes TraceDet, a hallucination detection framework designed specifically for Diffusion Large Language Models (D-LLMs). The authors formulate the denoising process as an action trace and apply information bottleneck principles to identify the most informative sub-trace for detecting hallucinated outputs. Experiments on LLaDA-8B-Instruct and Dream-7B-Instruct across three question-answering datasets demonstrate an average AUROC improvement of 15.2% over baseline methods.
1) This work addresses the underexplored problem of hallucination detection in D-LLMs, where generation proceeds via multi-step denoising rather than single-step autoregression. To the reviewer's knowledge, existing hallucination detection techniques are primarily designed for Auto-Regressive LLMs (AR-LLMs) and fail to exploit the intermediate dynamics of D-LLM decoding. The approach is strongly motivated by identified literature gaps: existing AR-LLM detectors rely on single-step signals, which fail to capture D-LLM dynamics, as empirically evidenced by the paper's observations (Figure 1) and baseline underperformance.
3) The most significant concern relates to impact. While technically valuable, the community actively working on or using D-LLMs remains relatively limited; thus, the work's impact is currently more potential than immediate. Furthermore, the reliance on an "inspired by" claim for the IB principle (Section 3.3) represents a missed opportunity for deeper theoretical insight. Finally, by testing exclusively on QA datasets, the paper proves its concept in a controlled environment but does not address more challenging, unstructured tasks where hallucination detection is arguably most critical. These limitations collectively suggest that while the paper constitutes a solid contribution, its overall significance is not yet fully established.

**Strengths:**

1. The problem formulation is sound. Hallucination in the emerging class of D-LLMs is fundamentally different from that in AR-LLMs. The denoising process naturally introduces three distinct mistake patterns that classical methods for AR-LLMs cannot identify or methodically benefit from.
2. The methodology is innovative and intuitive. Leveraging the trajectory trace and extracting partial steps successfully aligns with the identified problem. The introduced Information Bottleneck (IB) principle naturally connects token-level entropy with uncertainty identification in a theoretical and principled manner. The correctness of the derived lower bound for optimization is carefully verified.
3. The evaluation evidence is sufficient and rigorous. The experimental results are convincing. The paper demonstrates consistent gains compared to output- and latent-based methods across comprehensive QA benchmarks. In addition to the main experiments, the ablation studies effectively demonstrate both the practical value and efficiency claims.

**Weaknesses:**

1. The assumptions underlying the IB principle application are not formally justified. The application of IB is presented more as an inspiration than a formally justified choice. The paper provides limited empirical evidence or theoretical citations to establish a solid foundation for its core methodology, which undermines the completeness of the research, despite the mathematical soundness of the IB derivation and promising results.
2. The scope is limited, and the impact should be further emphasized. While the paper aims to address hallucination detection, the experiments are confined to multiple-choice, open-ended, and contextual QA datasets. Hallucinations primarily occur in open-domain generations, whereas QA benchmarks are often considered closed-domain. The reviewer suggests evaluating hallucination mitigation for more open-ended and real-world generation tasks.

**Questions:**

1. What underlying causes drive the observed hallucination patterns? Can you provide analysis linking specific denoising dynamics to the emergence of hallucinations?
2. Could you elaborate on the key challenge that "the hallucination-relevant action may be sparse and unevenly distributed across the action trace, and not every action contributes equally to the emergence of hallucination"? Is there any theoretical or empirical support to justify the necessity of selecting a sub-trace A_sub?
3. How would TraceDet perform on long-form generation tasks (e.g., summarization, creative writing) where hallucinations are more prevalent and consequential? What adaptations would be required for such scenarios?
4. Is the assumption of independence for A_sub reasonable? Although Figure 5(b) shows the impact of different τ values, this choice remains questionable and warrants further justification.

---

> ### Author Response · Authors · 2025-11-20
> **Rebuttal by Authors (part 1)**
>
> We sincerely thank the reviewer for the detailed and encouraging assessment of our work. We appreciate the acknowledgement of the soundness of our problem formulation, the novelty and theoretical grounding of our methodology, and the rigor and sufficiency of our empirical evaluation. Your positive feedback is highly motivating, and we address your comments point by point below.
>
> ### **Q1: What underlying causes drive the observed hallucination patterns, and can you provide analysis linking specific denoising dynamics to their emergence?**
>
> **A1:** We appreciate the reviewer’s thoughtful question. Our current analysis of hallucination patterns is primarily empirical, aiming to characterize observable patterns in the denoising trajectory rather than establish a causal mechanism. At present, conducting a systematic evaluation of different denoising dynamics remains challenging due to the lack of standardized benchmarks and ground-truth annotations that can distinguish fine-grained hallucination behaviors across steps. While the underlying mechanisms behind these behaviors remain an open question, the patterns we observe reveal characteristics that differ from those of AR-LLMs.  These observations validate the denoising trajectory as a meaningful analytical space and justify the design of TraceDet in leveraging stepwise dynamics to detect hallucinations.
>
> Building on these observations, an important extension is to develop benchmarks and analytical tools that enable deeper examination of how specific denoising dynamics give rise to hallucinations.
>
> ### **Q2: Theoretical or empirical support to justify the necessity of selecting a sub-trace $A_{\text{sub}}$​.**
> **A2:** This question highlights an important design motivation of our method. In D-LLMs, the generation trajectory consists of multiple denoising and remasking steps. However, hallucination-inducing behaviors such as the emergence and eventual lock-in of hallucinated keywords can occur at various points along the trajectory. While some examples may exhibit errors that persist throughout all steps, in many observed cases hallucination signals appear unevenly across the denoising process (as illustrated in Figure 1 and Appendix F). Consequently, uniformly weighting all actions introduces noise, which weakens the discriminative signal for hallucination detection.
>
> However, there is no benchmark that labels which intermediate steps are truly hallucination-informative, and manually annotating such fine-grained subinstances is prohibitively laborious. In this context, the IB formulation provides a principled solution: it enables the extractor $g_\theta$ to retain only those actions with high mutual information with the hallucination label, effectively filtering out low-information or redundant steps. This selective compression aligns with classical IB theory [1] and improves both stability and generalization.
>
> Empirically, TraceDet trained with sub-trace selection consistently outperforms the variant w/o masking (trained on the full trace) in AUROC as shown in Table 2. This observation is further supported by extensive qualitative analysis across hundreds of D-LLM generations, where hallucination-related dynamics are found to be sparse and localized in time.
>
> In summary, selecting a sub-trace is not a heuristic simplification but a principled mechanism for isolating the truly informative denoising steps where hallucinations emerge.

---

> ### Author Response · Authors · 2025-11-20
> **Rebuttal by Authors (part 2)**
>
> ### **Q3: Scope and generalization to open-domain generation**
> **A2:**  Thanks for these constructive suggestions! In our manuscript, we evaluated TraceDet on multiple datasets, including open-ended QA, contextual reasoning, and multiple-choice tasks, which together cover a broad range of reasoning and factuality scenarios. These benchmarks were chosen because they provide well-controlled hallucination labels and reproducible evaluation pipelines, which remain highly challenging to obtain in open-ended tasks such as creative writing, where distinguishing creativity from hallucination is inherently subjective. Moreover, HotpotQA requires LLMs to read long contexts and produce summarization-style factual responses, which already serves as a proxy for long-form generation. Our experiments on these benchmarks demonstrate that TraceDet achieves consistent performance gains across all baselines.
>
> We agree that extending TraceDet to long-form generation tasks would provide additional insight into its practical impact. However, prior studies have also noted that current D-LLMs face significant challenges in producing coherent long-form text, which makes reliable hallucination detection in such settings inherently difficult [2]. Very recent work has begun exploring ways to strengthen the long-text generation capabilities of D-LLMs, but this area remains in an early stage and evaluation practices are still evolving [3, 4]. Given these limitations, we were unable to include large-scale open-domain long-form experiments within the time and computational budget of the rebuttal phase. Nevertheless, the action-trace formulation of TraceDet is not restricted by generation length and can be directly applied to such tasks. For long-form generation, one could adapt TraceDet by incorporating sentence-level segmentation and hierarchical temporal embeddings to capture document-level denoising dynamics.
>
> We appreciate this suggestion, as it clearly points to an exciting and meaningful direction for future work. We plan to extend TraceDet to open-domain and long-form generation benchmarks to further validate its generality and real-world applicability.
>
> ### **Q4: Is the assumption of independence for A_sub reasonable?**
> **A4:** This is a thoughtful question that strengthens the rigor of our modeling. The independence assumption for $A_{\text{sub}}$ is introduced mainly as a computational simplification to make the information bottleneck principle tractable during sub-trace selection. Such independent modeling has been widely adopted and theoretically justified in prior applications of the information bottleneck to time-series data and graph-based data [5, 6, 7].
>
> We would like to thank you again for your comments and acknowledgement!
>
> [1] Tishby et al. Deep learning and the information bottleneck principle 2015 ieee information theory workshop (itw)
>
> [2] Huang et al. Ctrldiff: Boosting large diffusion language models with dynamic block prediction and controllable generation.  arXiv preprint
>
> [3] He et al.  Ultrallada: Scaling the context length to 128k for diffusion large language models.  arXiv preprint
>
> [4] Liu et al. LongLLaDA: Unlocking Long Context Capabilities in Diffusion LLMs. AAAI 2026.
>
> [5] Liu et al. TimeX++: Learning Time-Series Explanations with Information Bottleneck. ICML 2024
>
> [6] Yu et al. Graph Information Bottleneck for Subgraph Recognition.  ICLR 2021
>
> [7]Chen et al. Learning Causally Invariant Representations for Out-of-Distribution Generalization on Graphs. NeurIPS 2022

---

### Official Review · Reviewer_h49A · 2025-10-31

**Soundness:** 3
**Presentation:** 3
**Contribution:** 3
**Rating:** 4
**Confidence:** 3

**Summary:**

The paper introduces an information bottleneck based approach (called TraceDet) for identifying hallucinations in diffusion large language models (D-LLMs). TraceDet relies on intermediate denoising steps generated during inference of a D-LLM to identify potential hallucinations in the outputs. On a high level, it works by first identifying a subset of most informative actions (predicted masked tokens) pertaining to the factuality of the output and predicts a binary label for hallucination. The paper compares TraceDet against existing approaches for hallucination including both output-based and latent space-based methods, showing favorable results.

**Strengths:**

- D-LLMs are an emerging area of research and practical interest (esp when compared to Autoregressive, transformer based models). The paper addresses the challenge of detecting hallucinations in D-LLM outputs, while leveraging the information from intermediate denoising steps during inference. This research direction is pretty novel, as far as I know.
- TraceDet is based on well established mathematical concepts (namely, information bottleneck). Though I haven't checked the formulation thoroughly, to the best of my knowledge it appears correct and I am happy to see methods with some theoretical grounding.
- The paper uses well established benchmarks in the area of hallucination detection, which makes judging the work easier.

**Weaknesses:**

- Please correct me if I am wrong, but don't many if not all of the baselines not involve any training of a hallucination detector? They are purely inference time, statistics based methods. I believe this makes the comparisons unfair. I am giving a weak reject for this reason alone.

**Minor Typos/Grammar** (these played no role in my score):
- There is an extra space after decimal point (15. 2%) in the abstract.
- In Section 3.1, do you mean something like "... remasking which retains the highest confidence tokens ..." instead of "... remasking which retains the most confidential tokens ..."?
- $s_t$ is missing TeX formatting in Section 3.2 -> Action.
- "d" should be capitalized in "d-LLM" in Section 3.2 -> Transition and in some other place in the manuscript.

**Questions:**

- If more comparisons with methods involving training a detector are provided, esp with similar training complexity, I am happy to raise the score. If not, please provide a justification for why such additional comparisons are unnecessary.
- Since IB is a form of regularization, can we please see comparison with other kinds of regularization? For example, one could add some L2-regularization in the TraceDet w/o Masking and see if that how that compares.

---

> ### Author Response · Authors · 2025-11-20
> **Rebuttal by Authors**
>
> We thank the reviewer for the thoughtful and encouraging comments. We appreciate the recognition of the emerging importance of D-LLMs, as well as the novelty and theoretical soundness of TraceDet. We are also grateful for the acknowledgement of our use of standard benchmarks to enable clear evaluation. Your positive assessment and careful reading are very encouraging to us. Our responses to your concerns are summarised as follows:
>
> ### **Q1: Comparison with respect to training baselines for fair comparison.**
> **A1:** We thank the reviewer for raising this point and appreciate the opportunity to clarify our evaluation setting. Our study includes training-based baselines, which have comparable training complexity to TraceDet. We realize that this might not have been clearly emphasized and have refined the revised version to describe these comparisons and architectural details more explicitly in the Appendix.
>
> TraceDet is a lightweight hallucination detector composed of a 2–5-layer Transformer and a 2-layer MLP, with a carefully designed architecture, training objective, and embedding strategy. It only requires intermediate generations from D-LLMs as input, without additional inference or re-activation of the original model, making it architecturally simple and efficient to train.
>
> In this work, we categorize existing hallucination detection approaches into two main families: output-based and latent-based methods. Most training-based methods fall into the latent-based category. We compare TraceDet against three representative latent-based detectors, including two training-required state-of-the-art methods (CCS and TSV, current sota). Both CCS and TSV require reactivation of the full D-LLM during training and inference, as they operate on latent representations of question–answer pairs. In practice, these methods incur comparable training costs. Additionally, we include one strong training baseline TraceDet w/o masking from our ablation study. For fairness, all training-based methods were trained on identical hardware, and their training times are reported in the supplementary material for transparency.
>
> To the best of our knowledge, training-based methods that operate solely on model outputs without external information sources are relatively underexplored. Following prior work in hallucination detection, we therefore include the four most commonly used and D-LLM–replicable output-based methods for comparison. Under this setting, TraceDet demonstrates clear advantages in both detection performance and inference speed.
> Furthermore, a significant body of training methods focuses on auto-regressive architectures, where classifiers operate on token-level latent states in real time [1]. While effective for AR-LLMs, such approaches cannot be directly applied to diffusion LLMs, where there is no one-to-one correspondence between latent states and output tokens. This distinction highlights the necessity of developing dedicated hallucination detection methods for diffusion-based LLMs, which is precisely the motivation behind TraceDet.
>
> ### **Q2: Using L2-regularization as an alternative.**
> **A2:** It is insightful to point out the regularization nature of TraceDet. TraceDet can indeed be viewed as a form of parameter regularization comparable to applying L2 regularization to TraceDet w/o Masking.
> In response to the reviewer’s suggestion, we conducted an additional experiment replacing our proposed $L_{\text{ext}}$ loss with an explicit L2 penalty. Both losses impose constraints on the cardinality of the extracted subset $A_{\text{sub}}$​; however, $L_{\text{ext}}$ additionally regularizes the mutual information between $A_{\text{sub}}$​ and $A$, thereby minimizing information loss during extraction. As shown in the table below, the proposed $L_{ext}$​ yields stronger regularization effects and better empirical performance than a direct L2 penalty.
>
>
>
> | Loss Type | TriviaQA (128) | TriviaQA (64) | HotpotQA (128) | HotpotQA (64) | CommonsenseQA (128) | CommonsenseQA (64) |
> |------------|----------------|----------------|----------------|----------------|----------------------|--------------------|
> | $L_{\text{ext}}$  | 78.1           | 86.7           | 75.1           | 76.0           | 84.7                | 84.1              |
> | L2 Loss    | 76.1           | 85.1           | 59.1           | 74.0           | 82.4                | 83.7              |
>
> Table: Comparison of AUROC (%) between Mask Loss and L2 Loss across datasets
>
> ### **Q3: Typo issue**
> **A3:** We thank the reviewer for pointing out the typo. We have corrected it in the revised version and carefully checked the manuscript for any similar formatting or spelling issues.
>
> [1] Su et al. Unsupervised Real-Time Hallucination Detection Based on the Internal States of Large Language Models. Findings of ACL, 2024.

---

> ### Author Response · Authors · 2025-11-29
> **Discussion**
>
> Dear Reviewer,
>
> Thanks for your time and effort during the review process. As the deadline of the rebuttal is approaching, we kindly ask if our responses address your concerns.
>
> Best wishes,
>
> Authors

---

### Official Review · Reviewer_FNRD · 2025-11-01

**Soundness:** 3
**Presentation:** 3
**Contribution:** 3
**Rating:** 6
**Confidence:** 4

**Summary:**

In this paper, the authors study the detection of hallucinations in text generations of Diffusion Large Language Models (DLLMs), as opposed to the majority of existing literature which strictly analyzes autoregressive LLMs. Motivating using an information bottleneck perspective, the paper introduces TraceDet, to identify the most informative and relevant action sub-trace during the denoising steps using a network g that predicts the mask, and another classifier f to detect hallucinated responses from truthful ones over the selected sub-sequences. Overall, TraceDet is shown to be effective on DLLMs on standard datasets such as TriviaQA, HotpotQA and CommonsenseQA.

**Strengths:**

1) The paper studies a highly pertinent problem of detection of hallucinations in DLLMs, and is the first to the best of my knowledge to provide a dedicated analysis. While this can potentially limit the applicable scope since most LLMs in use today are autoregressive, the findings of the paper have great scope toward future development and understanding of DLLMs.


2) The paper is well-written and introduces each key component in a clear and concise manner. The TraceDet framework introduced by the paper is motivated well, and analyses across the denoising trace rather than specific single-point failure modes that might occur. For instance, the representative D-LLM hallucination patterns such as Interleaving, Persistent errors and Inconsistent guesses help back the need for such trace-based analysis.


3) The proposed method TraceDet demonstrates consistent and notable performance gains over a comprehensive suite of existing baseline methods that were originally developed for autoregressive LLMs, including both output-based and latent-based techniques on standard datasets such as TriviaQA, HotpotQA and CommonsenseQA.


4) The paper also presents a fairly detailed analysis on the sensitivity of relevant hyperparameters such as the masking ratio $\tau$ and the regularization weight $\beta$, as well as re-masking strategies such as low-confidence,  entropy, random and top-k based masking. The latter further indicates that the proposed method would likely be effective in future variants of DLLMs as well, given the fairly consistent across different strategies.

**Weaknesses:**

1) MDP Formulation: The paper formulates the denoising process as a Markov Decision Process, however the proposed method does not adequately functionally utilise this MDP framework, given that the information bottleneck perspective is adopted. Further incorporation of MDP theory could potentially have been leveraged to help mitigate hallucinations, rather than focussing on detection alone.


2) Evaluation Metrics: The empirical evaluations rely solely on AUROC. In real-world applications (and in imbalanced datasets, which hallucination detection often is), AUROC when used in isolation can fail to adequately capture detection performance. Could the authors kindly provide other standard detection scores such as TPR at low FPR and the F1 score, to better analyze the practical efficacy of TraceDet?


3) The paper introduces an extractor $g_{\theta}$ and a predictor $f_{\phi}$ but provides limited detail on their architectural complexity. A more in-depth discussion of these components, along with an analysis of their training and inference costs relative to the main D-LLM, would be beneficial for assessing the method's practical efficiency. For instance, could these then be used to guide fine-tuning of the main DLLM to directly reduce generation of hallucinated responses?


4) The analysis in Section 4.3 with maximum token entropy did not appear to be clear - it  does not really help highlight how TraceDet is helping improve the discriminability, especially with Figure 3a. Furthermore, is it expected that the averaged trace entropy for truthful samples is in general higher than that for hallucinated samples?


5) The paper compares TraceDet to baselines adapted from autoregressive LLMs, many of which utilize multiple completed generations. A more direct and potentially stronger baseline would be to adapt these consistency-checking principles within the D-LLM's trace. For example, applying a consistency check across intermediate steps of a single generation would directly engage with the paper's core hypothesis.




Minor Typos:

Equation 1: $min_{f \in H}$ L(Y, h(r0)) , should likely be $min_{h \in H}$

Line 197: asked tokens from $st$ ---> asked tokens from $s_t$

Line 229 to 231: Minor grammatical errors

**Questions:**

Kindly refer to the questions mentioned in the weaknesses section above.

In Equation 1, the summation over i is quite unclear. This could be improved by providing more details in the main paper, as is currently given in Appendix D.

---

> ### Author Response · Authors · 2025-11-20
> **Rebuttal by Authors (part 1)**
>
> We really appreciate your time and insightful comments. Thank you so much for acknowledging the method novelty, concise paper writing, notable performance gain, and detailed analysis of our work. The response to your concerns is summarised as follows:
> ### **Q1: The confusion raised by MDP Formulation.**
> **A1:** We sincerely thank the reviewer for carefully examining our method definition and for highlighting this valuable point. An MDP is a structured framework that represents a stochastic decision process through states, actions, and their transition dynamics. Our use of the MDP formulation is primarily conceptual: it provides a clear and structured way to describe the State–Action–Transition dynamics of D-LLMs, while intentionally omitting unnecessary components such as the rewards.  This abstraction helps formalize the relationship between intermediate noisy sequences $ r_{T-t} $ and their progressively denoised counterparts $ \hat{r}_{T-t-1} $, which supports our definition of the action trace and the information they encode about potential hallucinations.
>
> We completely agree that the MDP formalism can go beyond structural description. As the reviewer insightfully suggests, incorporating reinforcement signals like reward-based remasking or corrective guidance could indeed enable hallucination mitigation, not just detection. We deeply appreciate this suggestion on extending TraceDet into an MDP-driven correction framework by penalizing diffusion steps that drift toward hallucinated regions of the trajectory. This is an exciting and promising direction, but due to the time constraints of the rebuttal phase, we plan to pursue it in future work.
> ### **Q2: Choice and Diversity of Evaluation Metrics.**
> **A2:** The reviewer’s suggestion on extending standard detection scores is very helpful for assessing the practical reliability of hallucination detection with TraceDet. In our submission, we primarily report AUROC, as it is the most widely used metric for evaluating hallucination detection [1, 2]. Using AUROC allows for fair and direct comparison with prior works, including Eigenscore and TSV, which also adopt AUROC as their main evaluation measure.
>
> We agree that additional metrics, such as F1 score and TPR at low FPR, provide complementary insights into real-world detection efficacy. The detailed comparison is provided in the following table. As shown in Table 1-2 below, TraceDet consistently achieves a higher F1 score and TPR@FPR=0.1 than baselines, demonstrating stronger hallucination detection efficiency.  Although TraceDet outperforms all baselines, its stability under very low FPR settings can be further improved, for example, through regeneration strategies or majority-voting aggregation.
>
> | Method  | TriviaQA (128) | TriviaQA (64) | HotpotQA (128) | HotpotQA (64) | CommonsenseQA (128) | CommonsenseQA (64) |
> |--|--|--|--|--|--|--|
> | Lexical Similarity | 0.499 | 0.506| 0.598 | 0.552 | 0.800 | 0.826 |
> | EigenScore| 0.560 | 0.518 | 0.639| 0.569| 0.798 | 0.829|
> | CCS | 0.593 | 0.650 | 0.473 | 0.530| 0.610| 0.535 |
> | TSV | 0.705 | 0.676 | 0.685  | 0.651 | 0.625| 0.576 |
> | TraceDet | 0.767  | 0.802| 0.738 | 0.687 | 0.897| 0.902 |
> #### Table 1: F1 scores comparison between TraceDet and baselines
>
> | Method             | TriviaQA (128) | TriviaQA (64) | HotpotQA (128) | HotpotQA (64) | CommonsenseQA (128) | CommonsenseQA (64) |
> |--|---|--|--|--|--|--|
> | Lexical Similarity | 0.145 | 0.156 | 0.198 | 0.193| 0.000 | 0.000|
> | EigenScore | 0.239  | 0.296 | 0.235 | 0.241| 0.402 | 0.268 |
> | CCS | 0.090 | 0.269 | 0.118 | 0.103| 0.142 | 0.195 |
> | TSV| 0.235 | 0.268| 0.141 | 0.161 | 0.236 | 0.146|
> | TraceDet | 0.414  | 0.596| 0.193 | 0.391| 0.615| 0.616|
> #### Table 2: TPR@FPR=0.1 comparison between TraceDet and baselines

---

> > ### Author Response · Authors · 2025-11-20
> > **Rebuttal by Authors (part 2)**
> >
> > ### Q3: Architectural complexity and Computational Efficiency of TraceDet. Finetuning D-LLMs with TraceDet
> >
> > **A3:**  We appreciate the reviewer’s insightful comments, and this is a valuable question that helps assess the practicality of TraceDet. We are happy to provide more implementation details and will include them in the revised appendix for clarity. Specifically, the extractor $g_\theta$ is implemented as a 1-layer Transformer and a linear module, while the predictor $f_\phi$ is a 2-layer MLP. A temporal embedding module (2–4-layer one-head Transformer) is applied before the subtrace extraction to inject positional/temporal information into intermediate steps, and its outputs are then consumed by the extractor and predictor. Together, these modules form a lightweight post-hoc detector, **with fewer than 10M parameters in total**. The table below shows that, when evaluated under the same hardware and batch size, TraceDet exhibits a comparable average epoch training time, introducing only a marginal overhead to D-LLM training.
> >
> > We also value the reviewer’s constructive suggestion regarding the use of these modules to guide fine-tuning of the base D-LLM for hallucination mitigation. While TraceDet currently focuses on detection, the gradient-level signals produced by $g_\theta$ and $f_\phi$ can naturally inform mitigation strategies. For example, they can be used to perform gradient-guided remasking during decoding or to reweight fine-tuning losses on hallucination-prone steps. This represents a promising direction to connect TraceDet with proactive hallucination reduction, which we plan to explore in future work.
> >
> > | Method   | Training Time (s) |
> > |---|----|
> > | CCS | 3.64 |
> > | TSV | 19.2 |
> > | TraceDet  | 2.25 |
> >
> > #### Table 3: Comparison of average epoch training time across TraceDet and train-based baseline methods on TriviaQA using 1700 training samples
> >
> > ### **Q4: Explanation of analysis in Section 4.3**
> > **A4:** We acknowledge this and have expanded our analysis to clarify the interpretation of Figure 3(a). The average maximum token entropy analysis aims to statistically evaluate how TraceDet improves consistency in subtrace extraction. As shown in Figure 3(a), TraceDet tends to regularize the entropy distribution, lowering both the mean and variance, which reflects more stable subtrace selection and consequently more consistent hallucination detection. This section offers empirical insight into why TraceDet works, rather than a theoretical explanation, highlighting its role in stabilizing the denoising trajectory.
> >
> > Regarding the observation that hallucinated outputs show lower average maximum token entropy than truthful ones, we would like to note that this behavior originates from the intrinsic denoising dynamics of D-LLMs, rather than from TraceDet. Existing research has suggested that hallucinated output can also result in a high entropy pattern [3]. While our current analysis focuses on statistical observation, we view this as an interesting direction for further theoretical exploration, which we will discuss more clearly in the revised version.
> >
> > ### **Q5: Semantic consistency as a potential baseline.**
> > **A5:** Stepwise semantic consistency in the denoising trajectory is indeed a natural signal for hallucination detection. Unlike AR-LLMs, D-LLMs generate via iterative denoising and remasking, exposing semantic cues at each step without the heavy resampling required by AR-based consistency checks.
> >
> > However, as illustrated in Figure 1 and Appendix F, the intermediate generations of D-LLMs are often less semantically meaningful, dominated by placeholder fragments and repetitive partial tokens such as “the the the ansanswer ans.” As a result, semantic-entropy or lexical-similarity metrics, which are designed for clean and fluent sentences, become largely ineffective for consistency alignment. The noisy intermediate space of diffusion decoding therefore requires more structured modeling to extract meaningful signals.
> >
> > Consistency-based methods could still be complementary, but adapting them to diffusion’s noisy intermediate states requires finer-grained modeling and additional method development, which is beyond the scope of the rebuttal period. We view this as an interesting direction for future exploration.
> >
> > ### **Q6: Equation formatting**
> > **A6:** We thank the reviewer for pointing out the unclear notation, which refers to Eq. (6). We have clarified the summation index and notation in the revised version and added a concise explanation in Appendix D for completeness.
> >
> > We would like to thank you again for your comments and acknowledgement!
> >
> > [1] Chen et al. INSIDE: LLMs' Internal States Retain the Power of Hallucination Detection. ICLR 2024
> >
> > [2] Park et al. Steer LLM Latents for Hallucination Detection. ICML 2025
> >
> > [3] Bai et al. Mitigating Hallucinations in Large Vision-Language Models by Adaptively Constraining Information Flow. AAAI 2025.

---

### Author Response · Authors · 2025-12-01
**Rebuttal Summary for Area Chair**

Dear Area Chair,

We sincerely appreciate your work during this challenging and unexpected review situation. Since the rebuttal period was unexpectedly interrupted and due to the time limit we did not have the opportunity to receive further responses from the reviewers, we would like to provide a brief summary about the key clarifications and improvements addressed in our rebuttal. All details are available in the full rebuttal.

**TraceDet** contributes a principled action-trace formulation for analyzing D-LLMs, an information bottleneck–based sub-trace selection mechanism, a lightweight detector with minimal computational overhead, and consistent improvements over both output-based and latent-based baselines across all evaluated settings. Across the three reviews, we are grateful for the reviewers’ acknowledgement of (1) the novelty of TraceDet, (2) the soundness of our methodological construction, (3) the clarity and conciseness of our writing, and (4) the strong empirical performance gains over competitive hallucination-detection baselines. In summary, we have incorporated all constructive suggestions into the manuscript, provided extensive new experimental evidence and clarified conceptual points raised by the reviewers.

For **Reviewer FNRD**, the main concerns focused on the modeling of the MDP, the need for additional evaluation, and proposed a consistency-based baseline for comparison. In response, we clarified that the MDP serves as a structural formulation for denoising trajectories, and we explained why consistency methods do not translate directly to the noisy intermediate states of D-LLMs. We provided additional experiments reporting F1 and TPR@FPR=0.1 across all datasets, where TraceDet consistently outperforms existing methods, and we provided training-time analyses demonstrating that TraceDet is lightweight and introduces minimal overhead.

For **Reviewer h49A**, the main concerns involved the number of training-based baselines included in our comparisons and the question of whether our IB-based loss could be replaced with simpler regularization such as L2. We clarified that our evaluation already includes multiple training-required latent-based detectors (CCS, TSV) with comparable computational cost, in addition to our own training-based ablation without the IB module. We also expanded the architectural and training details in the revised appendix to avoid possible misunderstanding. Following the reviewer’s suggestion, we implemented an L2-regularization variant; it consistently underperformed compared to our IB formulation, confirming the necessity of our proposed masking mechanism.

For **Reviewer 741H**, the main concerns centered on the justification for selecting a sub-trace and the applicability of TraceDet to long-form generation. We clarified that, since different denoising steps may vary in how much useful hallucination-related information they provide, the IB formulation offers a principled way to identify and retain the more informative steps without assuming all steps are equally effective. Regarding long-form applicability, we explained that although robust evaluation for long-text D-LLM generation is still emerging, TraceDet’s action-trace framework naturally extends to longer sequences and can be adapted through hierarchical or segment-level modeling.

We kindly ask the Area Chair to consider these clarifications, additional experiments, and manuscript improvements in the meta-review process. Thank you very much for your time and efforts.

Sincerely,

All authors

---

### Meta-Review · Area_Chair_wPC6 · 2025-12-26

**Summary:**

The paper proposes TraceDet for identifying hallucinations in diffusion large language models (D-LLMs). TraceDet relies on intermediate denoising steps generated during inference of a D-LLM to identify potential hallucinations in the outputs. Based on the feedback from reviewers, the decision was made to recommend it for acceptance. We congratulate the authors on their acceptance! On the other hand, authors should revise the paper taking into account the reviewers' comments, such as the issues and concerns mentioned in Weaknesses.

**Reviewer Concerns:**

Key concerns include (1) Evaluation Metrics (The empirical evaluations rely solely on AUROC), (2) incomplete theoretical justification of the information bottleneck and MDP formulation. Authors should revise the paper taking into account these and other concerns and comments.

**Reviewer Scores:**

Reviewers consistently rated soundness and presentation as good, with mixed assessments of contribution. Scores ranged from marginally below to marginally above the acceptance threshold, with multiple reviewers explicitly stating they would recommend for acceptance.

---

### Decision · Program_Chairs · 2026-01-26

Accept (Poster)